# Determining Hearing Thresholds in Dogs Using the Staircase Method

**DOI:** 10.3390/vetsci11020067

**Published:** 2024-02-02

**Authors:** Cécile Guérineau, Anna Broseghini, Miina Lõoke, Giulio Dehesh, Paolo Mongillo, Lieta Marinelli

**Affiliations:** 1Dipartimento di Biomedicina Comparata e Alimentazione, University of Padova, Viale dell’Università 16, 35020 Legnaro, Italy; cecilechantalcatherine.guerineau@studenti.unipd.it (C.G.); anna.broseghini@phd.unipd.it (A.B.); miina.looke@unipd.it (M.L.); lieta.marinelli@unipd.it (L.M.); 2Independent Researcher, Via Chiesanuova 139, 35136 Padova, Italy; giuliodehesh@gmail.com

**Keywords:** auditory perception, behavioral test, dog, hearing threshold, hearing sensitivity, staircase method

## Abstract

**Simple Summary:**

While several studies have focused on dogs’ behavioral and emotional responses to sound stimuli, very few have addressed basic features of hearing, such as sensitivity. The latter is commonly described using the minimal intensities a subject can perceive at different frequencies, called hearing thresholds. To the best of our knowledge, only one behavioral study has explored this parameter in dogs. To strengthen the current knowledge on dogs’ hearing abilities, we devised a behavioral testing procedure based on a staircase method, whereby the sound intensity assessed in one presentation is increased (i.e., made easier) if the dog failed in the previous presentation or decreased (made harder) if the dog succeeded in the previous presentation. In this way, dogs’ sensitivity is evaluated through multiple assessments around the actual threshold, which increases the reliability of the result. We used this method to determine hearing thresholds at three frequencies (0.5, 4.0, and 20.0 kHz), testing five dogs per frequency. The hearing thresholds were found to be 19.5 ± 2.8 dB SPL at 0.5 kHz, 14.5 ± 4.5 dB SPL at 4.0 kHz, and 8.5 ± 12.8 dB SPL at 20.0 kHz. No improvement in performance was visible across the procedure. The results show that the staircase method is a feasible and reliable approach for assessing hearing thresholds in dogs. They also suggest that dogs might be more sensitive to high-frequency sounds than previously thought. This could be the result of selective pressure linked to intraspecific communication needs and potentially aid in low-range communication.

**Abstract:**

There is a growing interest in performing playback experiments to understand which acoustical cues trigger specific behavioral/emotional responses in dogs. However, very limited studies have focused their attention on more basic aspects of hearing such as sensitivity, i.e., the identification of minimal intensity thresholds across different frequencies. Most previous studies relied on electrophysiological methods for audiograms for dogs, but these methods are considered less accurate than assessments based on behavioral responses. To our knowledge, only one study has established hearing thresholds using a behavioral assessment on four dogs but using a method that did not allow potential improvement throughout the sessions. In the present study, we devised an assessment procedure based on a staircase method. Implying the adaptation of the assessed intensity on the dogs’ performance, this approach grants several assessments around the actual hearing threshold of the animal, thereby increasing the reliability of the result. We used such a method to determine hearing thresholds at three frequencies (0.5, 4.0, and 20.0 kHz). Five dogs were tested in each frequency. The hearing thresholds were found to be 19.5 ± 2.8 dB SPL at 0.5 kHz, 14.0 ± 4.5 dB SPL at 4.0 kHz, and 8.5 ± 12.8 dB SPL at 20.0 kHz. No improvement in performance was visible across the procedure. While the thresholds at 0.5 and 4.0 kHz were in line with the previous literature, the threshold at 20 kHz was remarkably lower than expected. Dogs’ ability to produce vocalization beyond 20 kHz, potentially used in short-range communication, and the selective pressure linked to intraspecific communication in social canids are discussed as potential explanations for the sensitivity to higher frequencies.

## 1. Introduction

Differences in perceptual abilities among species are very common, and the range of frequencies of auditory stimuli to which animals are sensitive to is not an exception. Humans’ audible range spans from 20 Hz to 20 kHz, defining the limits of the infra- and ultrasonic range, respectively. Yet, some animal species are capable of hearing sounds in the infrasonic range [1], and others can perceive ultrasonic sounds, way above humans’ upper limit (for example whales [2], bats [3], foxes [4], dogs, and cats [5]). The hearing sensitivity of different species was shaped by many factors, such as the physical environment, exemplified by the evolutionary trade-off between sensitivity to high frequency and the production of specific calls in a forest environment [6] or the acoustic conditions affecting sound communication in air and underwater [7]; the species sociality, for instance, the more complex the social group, the higher the sensitivity to high frequencies in primates [8]; and the necessity to have a good sound localization acuity, i.e., animals with a wide visual streak and larger visual fields, usually prey, have a poorer sound localization ability compared to species with narrower fields, usually predators [9].

The present paper regards hearing capabilities in dogs. As of today, most of the experiments on dogs’ acoustic perception have focused on relatively complex aspects of sound processing, including the recognition, classification, and encoding of various acoustic stimuli. For instance, it was shown that dogs were able to represent mentally a caller [10,11], understand referential aspects of growls [12] or barks [13], express an empathy-like response [14], or extract vocal emotional information [15]. It was also shown that basic acoustic parameters are fundamental for dogs to understand different contexts of calling patterns [16] or to extract emotional information from a vocalization [17]. For example, the fundamental frequency and the minimum and maximum frequency used within a vocalization are crucial to appropriately understand information regarding the meaning, the caller properties, and the valence of a vocalization [15,16]. Yet, despite their importance for higher sound processing, basic aspects of dogs’ auditory perception, primarily including the sensitivity across the hearing range, have received much less scientific attention.

The capability of a subject to perceive sounds is usually represented through audiograms, i.e., graphical representations in which the minimal perceivable intensity of a pure tone is plotted against a range of frequencies. Dogs’ audiograms have been generated through electrophysiological methods such as BAER (brainstem auditory evoked response), often also referred to as ABR (auditory brainstem response), BAEP (brainstem auditory evoked potential), and others (see for instance [17,18,19,20,21]). Briefly, these methods entail the detection of neural electrical responses at the level of the brainstem, upon exposure to sounds of known frequency and intensity. It is possible, therefore, that the hearing range of dogs might have been underestimated. In addition, hearing thresholds for a given frequency may vary according to the different electrophysiological procedure. For example, dogs’ reported thresholds at 4 kHz vary from 13 to 55 dB SPL [20,22]. In turn, while these studies tended to agree on the range of best hearing sensitivity of dogs, not all of them agreed on the specific frequency they perceive at best, which is reported to be 2 kHz [18] or 12 kHz [20]. Finally, due to their varied characteristics, these types of evaluations detect electrical activity evoked by a sound stimulus at the level of the brainstem, not the cortex, thus not allowing for an evaluation of dogs’ conscious perception of sound. In fact, these physiological measures are primarily designed to assess the integrity of the auditory pathway in clinical evaluations rather than to assess perception. 

As opposed to neurophysiological approaches, psychoacoustical behavioral methods can assess the entire sound perception process, from primary auditory receptors to higher-order cognitive centers [23]. Moreover, behavioral methods have other advantages: comparative studies assessing the difference between physiological and behavioral measures of hearing sensitivity in humans reported higher and more variable responses across subjects when using ABR compared to behavioral methods [24]. In dogs, Markessis and colleagues [25] tried to compare electrophysiological thresholds with the behavioral thresholds obtained by Heffner [26], showing an important difference in the low-frequency threshold, with the behavioral threshold being twice as low as those obtained with the physiological assessment. Moreover, thresholds in the latter were slightly higher in all the other frequencies used. Furthermore, in physiological studies in dogs, the inter-subject variability can be as high as 25 dB SPL [20], while computed data regarding the inter-subject variability from Heffner [26] found a standard deviation of 12 dB SPL at 0.5 kHz. In summary, behavioral assessment seems to grant a better accuracy and stability of thresholds in both humans and dogs. 

Despite these advantages, very little emphasis has been placed on devising behavioral methodologies to assess fundamental aspects of dogs’ hearing. A handful of studies used a behavioral approach to study dogs’ ability to localize sound [27,28,29], but, to the best of our knowledge, only Heffner [26] performed a study using a behavioral procedure to assess hearing threshold. The latter identified the hearing range of four dogs as ranging from 63 Hz to 47 kHz, determining the endpoints when sounds were only perceived if above 60 dB SPL. The high-level sensitivity range of dogs in Heffner’s study, defined as the interval of frequencies in which sounds could be perceived, even if lower than 10 dB SPL, ranged from 4 kHz to 16 kHz, with the best sensitivity at 8 kHz. Nevertheless, the study does not report relevant information, for instance whether any improvement occurred throughout the assessment, which implied numerous exposures to the sound stimuli. However, in a sensory discrimination task, repeated exposure can improve the ability of the animal to distinguish stimuli, and assessing thresholds with an insufficient number of trials could lead to an underestimation of the actual threshold.

In this study, we decided to adopt the staircase method [30], also referred to as the “method of up and downs”, to determine dogs’ hearing thresholds. The main characteristic of this procedure is that the intensity of the stimuli over subsequent assessments is set based on the previous performance of the animal, i.e., it is increased if the animal fails to detect the sound or decreased if it succeeds. As a result, in most of the assessments, the intensity of the stimulus is set around the actual hearing threshold of the animal, which allows for a better estimation accuracy. Furthermore, the procedure generally involves both descending and ascending assessment for each subject. In the descending assessment, the threshold is reached by starting from an intensity level clearly above the threshold and progressively decreasing it; vice versa, in the ascending assessment, the threshold is reached by increasing the intensity after starting from an intensity level clearly below the threshold. This leads to a more complete evaluation, as the subjects’ performance in a sensory discrimination task may change depending on whether trials progress from easy to hard (descending) or vice versa (ascending). Finally, the staircase method also has the advantage of observing the performance of the subject around the hearing threshold across time, with the possibility to see a potential improvement. Therefore, the aim of this study was to assess the feasibility of the staircase method to assess hearing perception in dogs, to overcome the limits of the previous methods, and to broaden the current knowledge regarding dogs’ hearing abilities. 

## 2. Materials and Methods

Hearing thresholds were determined with a two-alternative forced-choice paradigm, using pure-tone sound at three frequencies. The frequencies were chosen in order to cover a wide range of the dogs’ audible spectrum, namely a low (0.5 kHz), intermediate (4.0 kHz), and high (20.0 kHz) frequency. 

### 2.1. Subjects

The sample consisted of nine pet dogs (six females and three males), 3.2 ± 2.2 years old, of various breeds and of different morphological types (Table 1). Five of the nine dogs underwent the assessment for more than one frequency; so, for each frequency, thresholds were obtained from five dogs. 

Dogs were recruited opportunistically, with the inclusion criterion of being between 1.0 and 7.9 years old, to avoid youthful exuberance and potential age-related hearing alterations [20]. Further criteria included having a good health condition, without known hearing impairments, being willing to cooperate in the laboratory setting, and having a high motivation for food. The dog owners were recruited on a voluntary basis and were all part of the University of Padova staff or students. More details regarding individual dogs’ characteristics and frequencies on which thresholds were assessed are presented in Table 1.

### 2.2. Experimental Setting

Experiments were conducted in a room (5.8 m × 4.7 m) equipped to reduce sound reflection and attenuate outside noise. Details of the setting (Figure 1) and the apparatus (Figure 2) have been described in detail previously [29]. Briefly, the apparatus consisted of a wire-mesh structure, at the center of which a soft headrest was positioned. The height of the headrest varied according to the height of the dog’s head, so that the dog could naturally rest its head on it while standing. The apparatus had lateral walls which could be adjusted according to the width of the dog’s head, to ensure the latter was straight, i.e., perpendicular to apparatus main axis. Speakers (custom-built, with a frequency response of 45 Hz–22 kHz) were placed on the floor in a fixed position at both sides of the apparatus, at 148 cm distance and with a 30° angle from the headrest, and turned towards it. Speakers were powered by a custom-built amplifier, based on a TA2024 amplifier chip (frequency response range: 20 Hz–22 kHz). Two food dispensers (Treat & Train, Premier ^®^, PetSafe, Knoxville, TN, USA) were placed on either side of the apparatus, at 132.5 cm from the headrest.

During the experimental procedures, one experimenter, the dog operator, sat at 1.5 m from the apparatus. Another experimenter, the sound operator, sat behind the apparatus, concealed from the dog’s view by a panel covered with sound-absorbing material.

All the experimental sessions were recorded with three cameras (two WV-CS570 and one WV-CP310, Panasonic, Delhi, India), recording details of the position of the dog’s head in the apparatus, the sound operator’s area, and the entire width of the room with the two food dispensers.

### 2.3. Acoustic Stimuli

Sounds at 0.5, 4.0, and 20.0 kHz were used throughout this study. Regardless of the frequency and phase of the protocol, the stimulus had a duration of 750 ms, with 250 ms fade-in and 250 ms fade-out, to avoid switch-transient phenomena. The sounds were generated with the Audacity^®^ software [31] and reproduced during the experimental procedures by a MacBook Pro Laptop (Apple Inc., Cupertino, CA, USA), which sent the sound to the amplifier and, in turn, to the appropriate speaker. 

A sonometer (2250-S, Brüel and Kjaer, equipped with microphones (Model Brüel and Kjaer 4144, Naerum, Denmark, for frequencies under 8 kHz and Model G.R.A.S. 40 AG ½”, Holte, Denmark, for 20 kHz)) was used to ensure that stimuli reached the headrest with the required intensities in dB SPL (re 20 µPa) and was calibrated with a Brüel and Kjaer calibrator (Type 4231).

The minimal intensity for each frequency was determined according to the ambient background noise level and was equal to 17 dB SPL at 0.5 kHz, 4 dB SPL at 4.0 kHz, and 2 dB SPL at 20.0 kHz using 1/3-octave bandwidth windows with a slow time weighting of one second. To convert the meter readings into free-field SPL, free-field correction curves for 0° sound incidence were used (product data manual of 1” pressure-field microphone Type 4144, Brüel and Kjaer; product data manual of 40 AG ½” ext. polarized pressure microphone Type 40 AG, G.R.A.S, available here https://www.grasacoustics.com/products/measurement-microphone-cartridge/externally-polarized-cartridges-200-v/product/167-40ag (accessed on 1 September 2023). 

### 2.4. Experimental Procedure

Determination of the threshold at any given frequency involved a three-phase procedure, consisting of preliminary training, training, and test phases; the latter was composed of two assessments, i.e., descending and ascending assessments, as described below.

After the dog’s familiarization with the setting and the experimenter, preliminary training was performed as described by Guérineau and colleagues [29]. Briefly, the dog was taught to place its head on the headrest upon a verbal and/or gestural cue and wait in a standing position. This was achieved using a shaping procedure and food as a primary reinforcer. In separate moments, the dog was also trained to associate the reproduction of a sound by a speaker with the possibility of obtaining food from the food dispenser next to it. Only one speaker was used in this phase. Eventually, the two steps were combined, so that dogs learned to wait at the headrest until a sound was produced and then reach towards the food dispenser to obtain a food reward. 

When the preliminary training was completed, the training phase began. This consisted of sessions of 24 trials in which dogs were presented with a sound at 70 dB SPL, at the same frequency which would later be assessed in the test phase. In each trial, the dog placed its head on the headrest following the cue by the dog operator. As soon as the dog’s head was in the correct position, the sound was produced by either speaker, and the dog could then approach one of the two food dispensers. If the dog reached towards the correct food dispenser (i.e., that next to the speaker which had reproduced the sound), it was rewarded by food and verbal praise (“Bravo!”). If the choice was incorrect (i.e., the dog chose the dispenser next to the speaker not reproducing the sound) or if the dog moved before a sound was produced, no food was delivered and the dog operator said “No” and recalled the dog to get ready for the next trial. The side where the sound was reproduced was randomized, with the constraint that sound could not be presented on the same side for more than three times in a row, and counterbalanced among the 24 trials of the session. The training phase ended when the dog made less than 3 mistakes in two consecutive sessions (>91% accuracy). Upon reaching such a criterion, the test phase began.

Test sessions were composed of a sequence of 24 trials, 14 of which were training trials and 10 of which were test trials. The procedure for each trial was identical to what was described for the training phase. The intensity of the sound in training trials was always 70 dB SPL. All test trials of one session had the same intensity, but across sessions, the intensity varied as described in detail below. The sequence always started with two training trials, followed by the alternation of one test and one training trial. The inclusion of training trials in the test sessions had several purposes: to hinder potential frustration and to ascertain that the dog was performing with appropriate attention/motivation and thus preventing the presence of false alarms (i.e., the dog starting to move before any sound sent). If the dog started to move before any production of sound, the ongoing trial would have been considered as a failure. If the dog failed more than 15% of the time within the training trials (i.e., 2 trials out of 14), the entire session was considered invalid, and data scored in that session were removed from the final dataset.

The side of the sound was counterbalanced within both training and test trials of the session and randomized as described for the training phase. If the dog showed any sign of distress or fatigue within one session, the session was stopped and repeated later. 

The procedure described above was used for two subsequent assessments, namely the descending and ascending assessments, in this order. For these assessments, a staircase method was applied for the determination of the sound intensity to be used in test trials, which was based on the dog’s performance in the previous session: if the dog failed in less than two test trials of the session, the latter was considered successful and a lower intensity would be used for the test trials of the next session. If the dog failed on three or more test trials, the test session was considered as failed and a higher intensity would be used for the test trials of the subsequent session.

The descending assessment started with sounds in test trials at 60 dB SPL; in the following sessions, the intensity was reduced with decrements of 10 dB SPL until the first failed session. The sessions after failure had 5 dB SPL increments each until the next successful session, which was followed by 3 dB SPL decrements in the subsequent sessions until the next failed session. The latter failure was considered as the starting point for the actual staircase procedure. Starting from this value, all subsequent changes in intensity were conducted in steps of 3 dB SPL. Sessions in which the performance of the dog reversed (i.e., the first failed session after a successful one or the first successful session after a failed one) were considered as reversal sessions. Sessions proceeded until a minimum of ten reversal points were observed and the last two reversals were within a range of 6 dB SPL.

Once the descending assessment was completed, the dog underwent the ascending assessment, which started from an intensity level which was, for any given dog, 10 dB SPL lower than the average of the six last reversals of the descending assessment for that dog. The increment in the intensity level for the subsequent sessions was conducted in steps of 10 dB SPL until the first successful session, then a decrement in steps of 5 dB SPL until the next failed session, and then an increment in steps of 3 dB SPL until next successful session. The latter success was considered as the starting point for the actual staircase procedure. All other procedural aspects were identical to those described for the descending assessment.

### 2.5. Data Collection and Analysis

The average intensity of the last six reversal sessions of both the descending and the ascending assessment was calculated. The latter was considered as the dog’s intensity threshold at any given frequency. 

Behavioral data were collected from videos using Observer XT software (version 12.5, Noldus, Groeningen, The Netherlands). A continuous focal animal-sampling procedure was used. For each trial, we collected the sound intensity level, the exact moment and the side the sound was produced, the moment when the dog made a choice (i.e., arrived within 45 cm of either food dispenser), and if the choice was correct. Collected data were used to calculate latency from the moment the sound was produced to the moment the dog made a choice. For the analyses, the mean latency of test trials and of training trials was calculated for each reversal session. The mean latencies were analyzed in a sub-sample of dogs, homogeneous for experience (i.e., all at their first assessment) and frequency, which included the five dogs undergoing the assessment at 4.0 kHz as their first assessment. To assess inter-observer reliability, two coders collected data from 25% of the 424 videos chosen so that all dogs and failure and success sessions were equally represented. The collected data were compared using the intraclass correlation coefficient, resulting in an inter-observer reliability of 0.82.

In order to determine whether there was an improvement in the dogs’ hearing threshold between descending and ascending assessments, we used a generalized estimating equations (GEEs) model. Separate models were obtained for the different frequencies. The dependent variable was the intensity level of the last six reversals in both descending and ascending assessments. The model included the type of assessment (descending or ascending) as a fixed factor and the dog’s name as the random factor accounting for the repeated measurement taken from the same dog. 

In order to see if experience influenced the overall performance of dogs who underwent the procedure for more than one frequency, the number of sessions needed to reach the first of the last six reversals was compared between the descending assessment within their first and second procedure using a Wilcoxon test.

A GEEs model was used to determine whether there was any change in latency across the procedure and whether the latency depended on the potential improvement across sessions. The model included the mean latency as the dependent variable. It included the type of assessment (descending/ascending), type of trial (training/testing), type of reversal (success/failure), and first- and second-level interactions as fixed factors and the dog’s name as a random term. A backwards elimination procedure was applied to obtain the final model. Post hoc pairwise comparisons with sequential Bonferroni correction were performed if a significant effect was found for any interaction or model term. 

Spearman correlations were computed for each frequency to determine if the dogs’ interaural distance and the hearing thresholds were correlated.

## 3. Results

At 0.5 kHz, dogs needed a mean ± sd of 18.4 ± 1.7 and 16.8 ± 3.0 sessions to complete the descending and the ascending assessments, respectively. At 4.0 kHz, dogs needed 23.6 ± 4.1 sessions in the descending assessment and 22.6 ± 7.0 sessions in the ascending assessment. At 20.0 kHz, dog needed 18.2 ± 2.2 sessions in the descending assessment and 12.6 ± 2.3 sessions in the ascending assessment.

Figure 3 shows the mean ± SD of the intensity threshold in both the descending and ascending assessments for the three frequencies. 

The hearing threshold was 19.5 ± 2.8 dB SPL at 0.5 kHz, 14.5 ± 4.5 dB SPL at 4 kHz, and 8.5 ± 12.8 dB SPL at 20 kHz. At 20.0 kHz, the large standard deviation is due to the fact that in the ascending assessment one dog performed at 28.5 ± 1.6 dB SPL, while the other four succeeded at the minimum intensity sent by our equipment (mean intensity ± SD = −2.8 ± 0.8 dB SPL). The GEEs model did not result in a significant effect of the type of assessment on dogs’ hearing threshold at 0.5 kHz (Wald X2 = 0.040, df = 1, *p* = 0.842), at 4 kHz (Wald X2 = 2.895, df = 1, *p* = 0.089), and at 20 kHz (Wald X2 = 2.083, df = 1, *p* = 0.149), indicating that a stable final threshold was already achieved at the end of the descending procedure.

Among the five dogs that performed more than two procedures, the number of sessions needed to reach the first reversal of the last six was 15.4 ± 2.8 sessions within their first descending assessment and 12.8 ± 1.1 sessions within their second descending assessment, with no significant difference (Z = 1.500, *p* = 0.104), indicating that there was no effect of experience regarding the speed to reach their threshold area. 

Spearman correlations between the interaural distance and the thresholds were not significant for any of the three frequencies (0.5 kHz: r = 0.526, df = 3, *p* = 0.362; 4 kHz: r = −0.200, *p* = 0.747; 20 kHz: r = 0.671, *p* = 0.216).

Table 2 reports the mean threshold obtained in dogs in the present study, in a behavioral study [26], and in a physiological study focused on the BAER [20] for comparative purposes. Table 3 reports individual thresholds from our study and Heffner’s [26] study for statistical comparison purposes. 

Regarding the latency to perform a choice, the interaction between the type of trial (training/testing) and the type of reversal (success/failure) was significant (*p* = 0.029), but post hoc pairwise comparison showed no significant difference. No difference was found in the latency time within the training trials according to the type of reversal (failure = 1.22 ± 0.18 s; success = 1.21 ± 0.16 s; *p* = 0.847). No difference was found in the latency time according to the type of assessment (Wald X2 = 0.660, df = 1, *p* = 0.417). 

## 4. Discussion

In this experiment, we successfully devised a behavioral procedure, based on a staircase method, to determine dogs’ hearing thresholds at different frequencies. The average threshold in our sample was 19.5 ± 2.8 dB SPL at 0.5 kHz, 14.5 ± 4.5 dB SPL at 4 kHz, and 8.5 ± 12.8 dB SPL at 20 kHz. The procedure implied two subsequent determinations of the threshold at each frequency for each subject. No improvement was observed between the first and second assessment, implying that the maximal sensitivity was reached at the end of the first assessment.

Only the thresholds we obtained at 0.5 kHz and 4.0 kHz are directly comparable with the only other study employing a behavioral approach [26]. The thresholds found at 0.5 kHz were similar between the two studies; however, we found a difference of about 11 dB at 4 kHz. The threshold found in our dogs was higher compared to that reported by Heffner. There seem to be no evident methodological differences nor any overt characteristics of the samples of the two studies that could explain the difference. Considering the small sample size of both studies, it is possible that the difference reflects the normal variability within the dogs’ population. By way of comparison, a study on red foxes [4] reports a variability of about 14 dB SPL among three subjects, highlighting that a relatively large variability between individuals exists also in other canid species and appears to be a natural phenomenon.

However, another relevant aspect is that the individual thresholds of our subjects fell in a smaller range compared to Heffner’s [26]. A potential explanation stems from a crucial characteristic of the staircase procedure, i.e., that most of the assessment is performed with a sound intensity in close vicinity to the dogs’ threshold. This, in turn, might have resulted in more accurate estimations of the thresholds than the method adopted by Heffner. Another possibility which could explain the broader range reported by the latter is that the procedure used in such a study might not have allowed dogs to reach their maximal sensitivity before termination. Indeed, it is well known that in sensory discrimination tasks exposure may improve the ability of the animal to distinguish different kinds of stimuli [32,33,34], and this also applies to dogs in acoustic perception tasks [29]. In the present study, we looked for changes in performance across the procedure and found no evidence of improvement, suggesting that the number of trials dogs were exposed to when they reached the first threshold (180 to 250) was sufficient to reach their maximal sensitivity. However, the same might not be true for the study by Heffner, which reports that a minimum of 120 trials were performed but does not mention whether any dog needed more than this or any assessment of changes in performance across the procedure. Since learning might not occur with the same efficiency in different subjects, it is possible that some dogs in Heffner’s study might have already reached their best performance at the end of the assessment, while other dogs’ thresholds could have improved if testing had continued for longer. This, in turn, might explain the larger variability found in Heffner’s study compared to the present study.

No direct comparison can be performed with previous studies for the threshold at 20 kHz as neither behavioral [26] nor electrophysiological studies [18,19,25] assessed dogs’ hearing thresholds at such frequency. Yet, the thresholds of our dogs at 20 kHz were similar to those reported by Heffner at 16 kHz. Based on both Heffner’s study and the electrophysiological audiogram, thresholds are expected to rise well before 20 kHz; therefore, this similarity was unexpected. Moreover, if we consider that the threshold of one of our dogs at 20 kHz was a likely outlier—potentially owing to hearing impairments which were not detectable during training and are unlikely to be representative of the dogs’ population—the average hearing threshold at 20 kHz would be lower, making the result even more relevant. It is also notable that in our study thresholds found at 20 kHz were lower than those found at 4 kHz, by about 6 dB SPL (12 dB if not considering the outlier). By comparison, Heffner [26] found similar thresholds at both 4 and 16 kHz. As already mentioned, due to a small sample size, these results cannot be considered conclusive, and it is possible that our dogs’ higher sensitivity at higher frequencies does not reflect a population-wide characteristic. Yet, if the outlier is excluded, our dogs’ thresholds at 20 kHz showed an extremely small variability, suggesting that this is not a spurious result.

It seems difficult to attribute our dogs’ lower thresholds to methodological aspects; for instance, ambient noise may sometimes mask the actual hearing threshold of animals [35]. Nevertheless, the ambient noise in our study had a lower intensity than the thresholds found at both 4 and 20 kHz and was therefore unlikely to have biased the results. 

One possibility is that dogs’ optimal hearing range extends to further than previously thought and to at least 20 kHz. In fact, the frequency at which dogs are most sensitive is still a matter of debate; for instance, electrophysiological studies found the highest sensitivity at frequencies as diverse as 2 kHz [18], 8 kHz [25], or 12 kHz [20]. Comparison with electrophysiological studies must be treated carefully, as some of them did not assess dogs’ responses throughout the entire hearing range. Nonetheless, the trend whereby the sensitivity at frequencies in the upper part of the range (e.g., 16 kHz) is better than at 4 kHz is supported by some studies [19,20].

That dogs would be highly sensitive to high frequencies should not be surprising. Ecological factors and, more specifically, selective pressures linked to intraspecific communication are believed to have shaped dogs’ hearing sensitivity towards higher frequencies. Indeed, there seems to be a relationship between canid species’ sociality and the use of high frequencies in communication. For instance, the combination of both low- and ultra-high frequencies within one vocalization allows for quick individual recognition and facilitates the localization of the caller [36]. This type of vocalization is common in dholes, a highly social and cooperative canid species, and, accordingly, dholes’ hearing range extends toward higher frequencies. By contrast, peak hearing sensitivity in non-obligatory social canids, such as foxes, where recognition and localization of conspecifics is less crucial to survival, seems shifted towards lower frequencies [37]. Dogs are also able to produce bi- and poly-phonations, including ultra-high frequencies [38], whose perception would benefit from an extended sensitivity in a broader range of frequencies. Moreover, as high frequencies are subjected to bigger attenuation [39], they can be heard only at a very close range; hence, they are used for short-range communication while avoiding eavesdropping from other group members. Sibiryakova and colleagues [38] investigated acoustical characteristics of whines in dogs and found ultra-high fundamental frequency that could reach 23 kHz. Like the previously cited studies, they suggested that ultra-high frequencies would be used for “tête-à-tête” conversation.

In the present study, there were no differences in the accuracy, nor in the latency to perform a choice, between the first, descending assessment and the second, ascending assessment. Therefore, no improvement occurred throughout the procedure, suggesting that the first assessment would have been sufficient to conclusively determine the dogs’ thresholds at any given frequency. The lack of significant improvement across the procedure is seemingly in contrast with a previous study of our group, where the staircase method was applied for the determination of sound localization abilities in dogs [29] and where a clear improvement was observed throughout the procedure. However, sound detection and sound localization are processed in separate regions of the auditory cortex, with localization involving a crucial role of non-primary auditory areas [40]. Due to the involvement of higher cortical areas, it is possible that sound localization maximum abilities require more experience than the relatively simpler process of sound detection. In humans, hearing thresholds did not improve across sessions and did not differ across methods [41], suggesting that maximal sensitivity is reached early on in this kind of task. Improvement through repeated exposure has been reported in humans performing minimal audible angle tasks [42] and other sound-localization-related tests [43]. Another potential explanation relates to methodological differences between our two studies. The perception of pure tones occurs earlier in the auditory stream and is less demanding in terms of cortical processing than that of more complex sounds [44]. It is then possible that the use of pure tones in the present experiment resulted in an easier task and, hence, a faster reaching of the maximal sensitivity and no evident improvement throughout the procedure than the white noise used in the sound localization study.

## 5. Conclusions

Taken together, the results indicate that the staircase procedure we devised is a feasible approach for the assessment of the minimal hearing thresholds in dogs. The results highlight the absence of improvement over time, leading to a reliable final estimation of the threshold. Relying on a simple discrimination task, the procedure appears to be easily applicable to most dogs, provided they are motivated by food. Even though the actual procedure used in this study is time-consuming, the lack of improvement emphasizes the fact that a reliable estimation could be obtained with the sole descending assessment, considerably shortening the time demand.

Overall, the thresholds obtained with this procedure are not dissimilar from those reported by the only other comparable study found in the literature. However, neither the present study alone nor the pooling of the two studies can be considered truly representative of the canine population. Therefore, the results prompt the application of the present methodology to larger-scale studies to obtain a more comprehensive representation of dog hearing abilities. One aspect that seems to deserve specific attention is the high-frequency domain, as the present results suggest that the range of optimal hearing in dogs might extend to higher frequencies than previously thought. 

## Figures and Tables

**Figure 1 vetsci-11-00067-f001:**
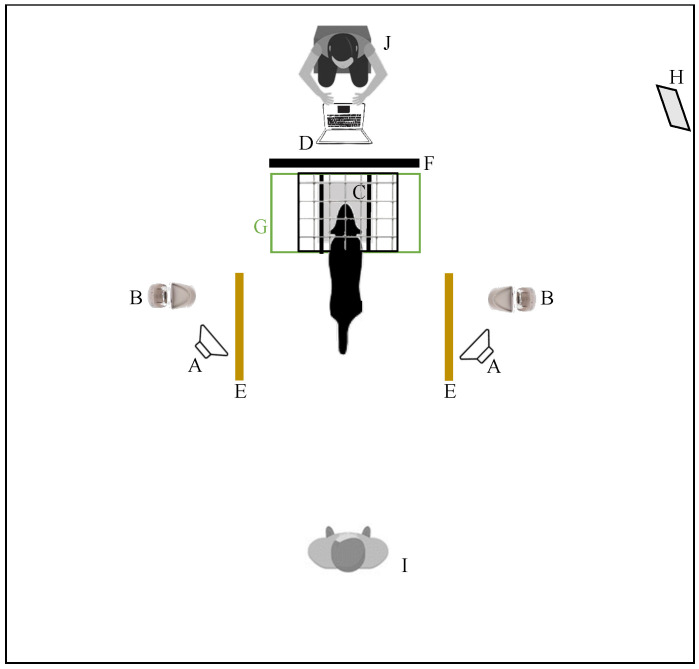
Schema of the room and spatial position of experimental elements seen from above. A, speakers; B, food dispensers; C, headrest; D, computer triggering sound; E, line defining dog’s choice; F, panel separating sound operator from dog’s view; G, apparatus; H, mirror; I, dog operator; J, sound operator.

**Figure 2 vetsci-11-00067-f002:**
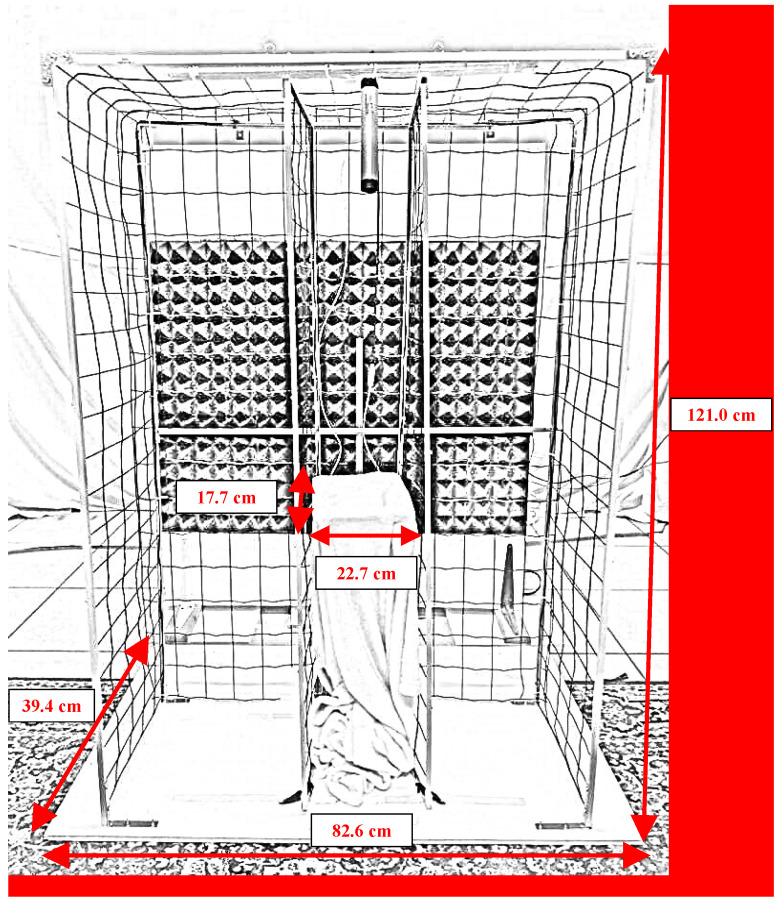
Detailed picture of the apparatus.

**Figure 3 vetsci-11-00067-f003:**
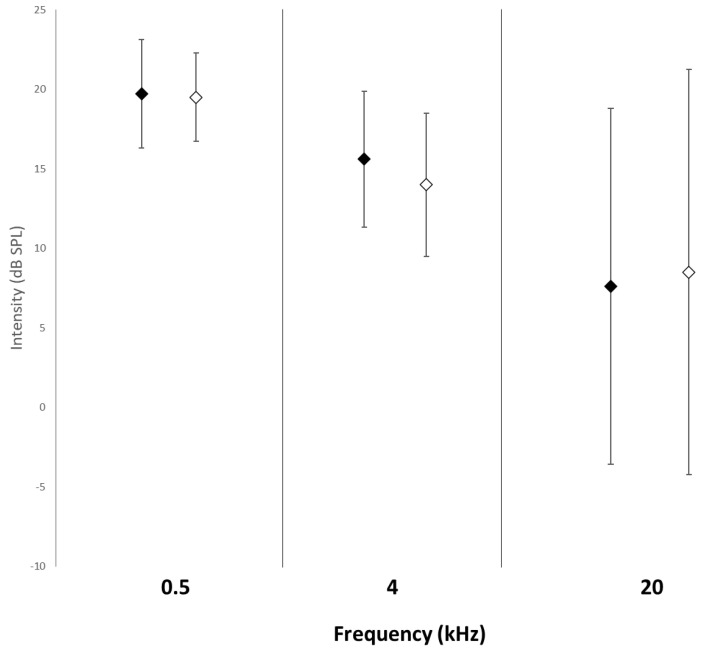
Mean ± SD of the intensity level of the six last reversals in both descending (black diamonds) and ascending (white diamonds) assessments for the three frequencies (kHz).

**Table 1 vetsci-11-00067-t001:** Demographics, ear shape, and interaural distance of the dogs and the frequency(ies) for which thresholds were assessed for each dog.

Dog	Age (y)	Sex and Reproductive Status	Breed	Ear Shape	Interaural Distance (cm)	Tested Frequency (kHz)
1	3.3	F/N	Mixed Breed	Not covering ^1^	12.3	0.5, 4 *, 20
2	7.8	F/N	Mixed breed	Covering ^3^	9.8	0.5, 4 *
3	1.1	F/I	Labrador	Partially covering ^2^	12.9	0.5, 4 *
4	1.2	M/I	Australian Sheperd	Not covering ^1^	15.0	0.5, 4 *
5	1.8	M/N	Mixed Breed	Not covering ^1^	12.9	0.5, 20 *
6	2.5	F/N	Golden Retriever	Partially Covering ^2^	13.0	4
7	3.1	M/I	Border Collie	Partially Covering ^2^	10.0	20
8	2.3	F/I	Mixed Breed	Not covering ^1^	12.0	20
9	5.5	F/N	Mixed Breed	Not covering ^1^	16.0	20

F, female; M, male; I, intact; N, neutered; ^1^ erect or semi-erect ears, not covering the canal entrance at all; ^2^ ears covering, without touching, the canal entrance; ^3^ ears covering and touching the canal entrance. * indicates the first frequency for which threshold was assessed, for dogs who were tested on more than one frequency.

**Table 2 vetsci-11-00067-t002:** Mean ± SD and range of thresholds (dB SPL) obtained by Heffner [26], the present study, and Ter Haar and collaborators [20] at 0.5, 4.0, and 16.0 or 20.0 kHz.

Study	Mean Dogs’ Age (y)	0.5 kHz Mean ± SD (Min–Max)	4 kHz Mean ± SD (Min–Max)	16 [1,3] or 20 [2] kHz Mean ± SD (Min–Max)
Heffner [1]	2.5	20 (11–24)	4 (−5–9)	6 (0–15)
The present study [2]	3.2	19.5 (17–21)	14.5 (10–17)	8.5 (2–34)
Ter Haar et al. [3]	1.9		13	12
	5.7		40	13

**Table 3 vetsci-11-00067-t003:** Individual hearing thresholds (in dB re 20 μPa) for five (present study) and four (Heffner’s study) dogs.

Frequency (in kHz)	Study	Individual Thresholds (in dB SPL)					Mean	Range ^3^	Mann–Whitney *p*-Value
0.5	Present	19.5	20.5	19.5	21.0	17.0	19.5	4.0	0.556
	Heffner	24	24	11	20		20	13	
4.0	Present	14.0	16.0	17.5	10.5	14.5	14.5	7.0	0.016
	Heffner	3	7	−5	9		4	14	
20.0	Present	2.0	2.0	2.0	3.0	33.5	8.5 ^1^/2.2 ^2^	31.5 ^1^/1.0 ^2^	N/A
16.0	Heffner	3	15	0	4		6	15	

^1^ With the outlier; ^2^ without the outlier; ^3^ difference between the maximal and the minimal value among individual thresholds; N/A: statistical analyses not performed due to different frequencies used among studies.

## Data Availability

The data are contained within the article.

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
