# Peer review of "Determining Hearing Thresholds in Dogs Using the Staircase Method"

_vetsci, 2024, doi:10.3390/vetsci11020067_

Round 1

Reviewer 1 Report

Comments and Suggestions for Authors

In this manuscript, the authors performed behavioural hearing tests at three frequencies in 9 dog individuals. The use positive reinforcement conditioning and a psychophysical staircase procedure to estimate the absolute hearing thresholds at 0.5,4, and 20 kHz. From their results, the authors conclude that dogs hear better than expected in the high frequency range and they attribute the use as of the staircase method as a possible cause.

While I very much welcome new behavioural audiogram data, especially in dogs where the effect of breed has certainly been understudied, I have some questions and concerns that I would ask the authors to address.

Major comments:

1.       My main question concerns the technical details of the stimulus characterization. For absolute measurements of sound pressure levels (SPL) it is important to indicate the reference level in µPa and the way in which the sound level meter was calibrated. Furthermore, the authors use a microphone which according to the manufacturers instructions is not suitable for measurements at 20 kHz. Even when applying a correction, how can the authors be sure that they do not underestimate the SPL of their 20 kHZ stimulus, which would be indistinguishable from the unexpectedly low threshold that they observed in their dogs? If the authors cannot be sure, I would suggest to either remeasure the sound levels or exclude the 20 kHz datapoints from the paper.

2.       Regarding the background noise: Please add some information about the frequency window size in which these where measurement and the time window about which the integration happened. Was a dB weighing used for the measurements? It would be good to provide the spectrum of the background noise.

3.       In psychophysics in is good practice to correct the hits (correct choices, accuracy reported in the manuscript) by the number of false alarms, i.e. the times the animal tried to guess. Did the authors assess false alarms? Given that the video data is available this should still be possible.

4.       The authors focus their comparisons only on the single behavioural audiogram by the Heffner group. However, other behavioural assessments of dog hearing have been published. I am attaching two examples and would ask the authors to include this in their discussion and comparative table.

5.       The threshold data in table 2 do not match the data from the paper. I suspect that this is because they have been estimated from the graph. I suggest taking the numbers from the following website which is hosted by the Heffners: https://www.utoledo.edu/al/psychology/research/psychobio/newaudiograms/dog2.html  I would also ask the authors to provide a similar table for the measurements in their paper. This will allow them to perform statistical tests to assess if their thresholds indeed are different or not from the earlier measurements by the Heffner group.

6.       I would appreciate if the authors could comment on why they did not test more frequencies to create a complete audiogram.

7.       Did the authors test for any differences between sexes or ear types?

Minor comments:

-          Line 54: Also dogs, cats and foxes here ultrasound

-          Line 58: Specify that this relationship was found in primates. Bats hear very high frequencies but many are not particularly social.

-          Lines 83-85: This sentence does not make much sense in the context of the paper as the authors are not aiming to fill the indicated research gap (outside of 0.5khz-32khz)

-          Lines 177/178: Please provide the frequency range of both speaker and amplifier

-          Line 250 and others: Please check the formatting, lines seem to be mixed up.

-          Table 2: Ter Haar is ref 17 or 18, but not ref3. I suggest to remove the NA to make the table easier to read

-          Lines 355-358 (and 395-399): The comparative interpretation of the thresholds should be supported by statistical comparison with the published data from the Heffner paper.

Reviewer 2 Report

Comments and Suggestions for Authors

This manuscript presents a procedure for testing hearing sensitivity in dogs using the staircase method. The authors include good justification for the use and success of this procedure and present sound scientific results. Aside from some minor wording edits, I have two suggestions for revision:

1. I see the explanation in Table 1 for the frequency thresholds each dog was tested on. However, I am not sure why some of them were only test on one and some tested on all three. Could you explain more about this?

2. Is there any concern for the age or breed/size of the dogs the may explain variability in their responses. I did not see any discussion about this and I am not sure if individual responses were compared beyond the examination for improvement across the trials. I think it could help to address this in the discussion.

Comments on the Quality of English Language

The paper is written very clearly with only a few minor edits needed. Here are a two places were edits could be considered:

Line 91: Sentence reads, "Finally, due to their very characteristics, these types..."-should this instead say "varied characteristics"?

Line 152: Sentence reads, "...went the assessment for more than one frequency, so for each frequency thresholds were...."-should this say "so for each frequency threshold"?

Reviewer 3 Report

Comments and Suggestions for Authors

Perhaps a paragraph discussing threshold estimation: its meaning, versus waveform resolution and pure-tone testing would help 

Round 2

Reviewer 1 Report

Comments and Suggestions for Authors

The authors did a good job with the revision.

There are only three further points I would ask them to comment on/address:

1.       Please specify which angle of incidence correction was used for the new microphone for measurements at 20 kHz: https://www.grasacoustics.com/products/measurement-microphone-cartridge/externally-polarized-cartridges-200-v/product/167-40ag

2.       This microphone has a thermal noise limit of 20 dBA.  Are the authors sure about their 2 dB SPL background levels at 20 kHz?

3.       Interpretation: The data does not provide support for the notion that the thresholds at 20kHz are lower than expected. They line up nicely with the values at 16 khz reported by Heffner et al. If anything, the values at 4kHz deviate from the previous report (as the statistics also show). I would therefore ask the authors to tone down the point on the 20 kHz difference.
